# MBR and QE Finetuning: Training-time Distillation of the Best and Most Expensive Decoding Methods

**Mara Finkelstein and Markus Freitag**
Google
{marafin,freitag}@google.com

## Abstract

Recent research in decoding methods for Natural Language Generation (NLG) tasks has shown that MAP decoding is not optimal, because model probabilities do not always align with human preferences. Stronger decoding methods, including Quality Estimation (QE) reranking and Minimum Bayes Risk (MBR) decoding, have since been proposed to mitigate the model-perplexity-vs-quality mismatch. While these decoding methods achieve state-of-the-art performance, they are prohibitively expensive to compute. In this work, we propose *MBR finetuning* and *QE finetuning*, which distill the quality gains from these decoding methods at training time, while using an efficient decoding algorithm at inference time. Using the canonical NLG task of Neural Machine Translation (NMT), we show that even with self-training, these finetuning methods significantly outperform the base model. Moreover, when using an external LLM as a teacher model, these finetuning methods outperform finetuning on human-generated references. These findings suggest new ways to leverage monolingual data to achieve improvements in model quality that are on par with, or even exceed, improvements from human-curated data, while maintaining maximum efficiency during decoding.

## 1 Introduction

Beam search and greedy decoding are the most common decoding methods used for Natural Language Generation (NLG) tasks, including neural machine translation (NMT). However, Eikema & Aziz (2020) showed that maximum *a posteriori* (MAP) decoding methods, which approximate the most likely prediction based on model probabilities, are suboptimal for NMT due to misaligned probability distributions. They instead proposed Minimum Bayes Risk (MBR) decoding as an alternative decoding method. Unlike MAP decoding, MBR decoding does not aim to produce the prediction (i.e. translation) with the highest estimated model probability. Instead, it chooses the prediction that is estimated to have the highest quality with respect to a utility metric. A follow-up study by Freitag et al. (2022a) showed that MBR decoding with neural utility metrics like *BLEURT* (Sellam et al., 2020) and *COMET* (Rei et al., 2020) significantly outperforms beam search decoding, according to expert-based human evaluation.

However, the main drawback of MBR decoding is that it is prohibitively expensive. In particular, the algorithm requires that, for every input query, a large number $n$ of candidates be generated from the model, and then an (expensive) scoring function be computed on every pair of distinct candidates $(n_i, n_j)$, for a total of $O(n^2)$ computations.

Given the significant quality improvements afforded by MBR decoding relative to beam search, we propose to distill the MBR quality gains at training time, without affecting decoding speed or resource usage. Despite its quality advantages, the slow inference speed of MBR decoding remains a limitation even when generating distillation data. As an alternative, we can rerank the same candidate model predictions using a neural quality estimation (QE) metric. Reranking is faster than MBR decoding, because its inference cost scales linearly with the number of candidate predictions, rather than quadratically. Fernandes et al. (2022) showed that reranking with neural metrics produces better predictions than beam search, and that it has similar benefits to MBR decoding.

In this work, we focus on the task of NMT, and show that we can benefit from the quality gains of MBR decoding and QE reranking by finetuning NMT models on MBR-decoded and QE-reranked outputs generated from monolingual sources, either via self-training or using an external teacher model, and then using a more efficient decoding method (such as beam search) at inference time.

Our contributions can be summarized as follows:

- We propose two finetuning methods, *MBR finetuning* and *QE finetuning*, each of which distills performance gains from MBR decoding and QE reranking, respectively, at training time while avoiding expensive decoding at inference time.
- Using the task of NMT, we show that these finetuning methods significantly outperform the base model across two language pairs, while finetuning on beam search output degrades quality.
- We show that both MBR and QE finetuning on top of a model finetuned on human translations yields additional quality improvements.
- We show that using a LLM as the teacher model substantially outperforms using a self-teacher for MBR and QE finetuning.
- Moreover, we show that both MBR and QE finetuning from the base student model using a LLM teacher even outperforms finetuning on human translations.

## 2 MBR DECODING AND QE RERANKING

Broadly, both MBR decoding and QE reranking can be decomposed into two steps:

1. Given a source segment, generate a list of candidate model outputs. In this work, we use sampling to generate the candidate translations from a NMT model.
2. Choose the best output based on a utility function. In this work, we use either a neural QE or a neural reference-based metric as utility function.

### 2.1 CANDIDATE LIST GENERATION

The first step is identical for both decoding strategies. We generate candidate translations using epsilon sampling (Hewitt et al., 2022) with $\epsilon$=0.02, which was shown to be the best sampling method for MBR decoding in Freitag et al. (2023).

### 2.2 MINIMUM BAYES' RISK (MBR) SCORING

MBR scoring uses the set $\mathcal{H}$ of samples obtained from the first step both as candidate translations and as "pseudo-references", then uses a reference-based utility metric to estimate the expected utility of each candidate translation with respect to the set of pseudo-references. The candidate with the highest expected utility is chosen as the best translation.

That is, given a utility metric $u(h, r)$ which estimates the quality of a candidate translation $h$ conditioned on a reference translation $r$, we select the Minimum Bayes Risk (MBR) translation $h^{mbr}$ from a set of hypotheses $\mathcal{H}$ as

$$h^{mbr} = \arg\max_{h \in \mathcal{H}} \frac{1}{|\mathcal{H}|} \sum_{y \in \mathcal{H}} u(h, y)$$

Freitag et al. (2022a) showed that neural utility metrics outperform lexical overlap-based metrics, so we use *BLEURT v0.2* (Sellam et al., 2020) as the utility function $u$. Note that the number of forward passes through the utility function required to compute $h^{mbr}$ is quadratic in the size of the candidate set $\mathcal{H}$. In practice, this means that MBR decoding is prohibitively expensive.

### 2.3 SCORING WITH A QE METRIC

Alternatively, QE reranking uses a reference-free utility metric to score the candidate translations, so that rather than requiring an average over all pseudo-references to compute each candidate's utility,

only a single forward pass through the metric model is required. Thus, QE reranking is linear, rather than quadratic, in the candidate size.

Formally, a QE utility metric $u(h, s)$ estimates the quality of a candidate translation $h$ conditioned on the source $s$, rather than on the reference. We select the best QE translation $h^{qe}$ of the source $s$ from a set of hypotheses $\mathcal{H}$ as

$$h^{qe} = \arg\max_{h \in \mathcal{H}} u(h, s)$$

There is no QE version of *BLEURT*, so we instead use *MetricX-XXL-QE* as our utility function. This metric has the same architecture and was trained on the same human judgements data as *MetricX-XXL*, the winning submission to the WMT'22 Metrics Shared Task (Freitag et al., 2022b). To make it a QE metric, we pass the source segment as input to the metric instead of the reference.

See Table 5 (Appendix A) for a meta-evaluation of *BLEURT* against *MetricX-XXL-QE* on the WMT'22 en→de MQM test set. According to both segment-level and system-level meta-evaluation metrics, *BLEURT* and *MetricX-XXL-QE* perform comparably. Also see Figure 1 for a comparison of the ranking of all sampled translations of a single source sentence by MBR scoring with *BLEURT* (as described in §2.2) versus QE scoring with *MetricX-XXL-QE* (as described in §2.3), and see Figure 2 for the MBR and QE score distributions of all sampled candidates versus the top-1 candidate.

## 3 MBR AND QE FINETUNING

MBR decoding and QE reranking are both inference-time solutions to the "model-perplexity-versus-quality mismatch" problem. We propose *MBR and QE finetuning* to instead ameliorate this mismatch at training time via direct adaptation of the model weights, without having to incur high costs and resource usage at inference time.

Concretely, the method of MBR finetuning can be decomposed into the following steps:

1. Dataset generation: Generate MBR-decoded translations (see §2.2) from some monolingual datasource using a teacher model $T$.
2. Finetuning: Finetune student model $S$ on the dataset generated in Step 1.
    2.1. *Variant 1: Self-training ($T = S$).* The finetuning is initialized from the same model checkpoint used to generate the MBR-decoded dataset.
    2.2. *Variant 2: Distillation ($T \neq S$).* The student model $S$ is finetuned on the dataset generated by the teacher model $T$.
3. At inference time, decode with the finetuned student model $S$ using an efficient decoding method such as beam search, greedy search, or sampling.

QE finetuning is analogous to MBR finetuning, with the only difference being that QE reranking (§2.3) is used as the decoding strategy instead of MBR decoding during dataset creation in Step 1.

## 4 RELATED WORK

MBR decoding with neural metrics was first proposed by Freitag et al. (2022a), where it was shown to outperform QE reranking when using *BLEURT v0.2* (Sellam et al., 2020) as the MBR utility metric and *COMET-QE-21* (Rei et al., 2021) as the QE utility metric. Fernandes et al. (2022) systematically explored different ranking strategies for so-called *quality-aware decoding* and also found that (top-1) QE reranking underperformed the beam search baseline. However, *tuned reranking*, where four different QE metrics are linearly combined with weights learned so as to maximize a reference-based metric on a validation set, showed strong improvements over beam search.

Shen et al. (2015) proposed a training-time approach to directly optimize evaluation metrics, which they call *Minimum Bayes Risk Training*. They initialized from a model already trained on the maximum likelihood estimation (MLE) objective and then used an alternative loss function which aligns the model's probability space with the evaluation metric. In particular, for each source sentence, they scored $N$ samples from the model using the evaluation metric, with the target side of the training example used as the reference. Then the risk for each sample was calculated by multiplying the

metric score with the sample's model probability, and the loss for this example was computed as the negative sum of the risks of all samples. So rather than using the single best sample according to the evaluation metric (as in MBR finetuning), they weighted the loss by the risk of all generated samples. Unlike MBR finetuning which uses a static teacher, their (self-)teacher evolves over the course of training as model weights are updated. They only considered *BLEU* (Papineni et al., 2002) as their evaluation metric (and did not consider neural metrics). Their method also requires references (unlike MBR finetuning), though using Bayes Risk scoring instead would be a natural alternative. The quality improvements achieved with this method were modest, and came at the cost of unstable and expensive training.

Gulcehre et al. (2023) proposed a method for offline reinforcement learning-based finetuning called *ReST*, which generates a finetuning dataset by sampling from the "policy" (i.e. model), scoring each sample with a QE metric, then iteratively finetuning on these samples using a reward-weighted loss based on the QE scores (after removing samples with scores below a fixed threshold, which becomes stricter at later iterations). The method of QE finetuning proposed in this paper, on the other hand, selects the top-1 sample based on QE score. Morevoer, *ReST* finetuning uses ancestral, rather than epsilon, sampling and samples from the original base training dataset, without exploring the effect of finetuning dataset used on model performance.

Extensive prior work in unsupervised and semi-supervised machine translation has also investigated how monolingual data can be used to improve translation model quality. The techniques of forward and backward translation have resulted from this line of research (Sennrich et al., 2016). A related line of work has focused on distilling large teacher models into smaller student models, by penalizing differences between the translations produced by the teacher and the student (Hinton et al., 2015). This work has shown that the quality and style of teacher models can be transferred to students (Freitag & Al-Onaizan, 2016). However, the use of alternative decoding algorithms to generate distillation data has not been previously explored, and distillation is traditionally performed using an external teacher model. This work addresses both of these gaps, and shows that self-distillation is effective (only) when the strongest and most expensive decoding algorithms are used.

## 5 EXPERIMENTAL SETUP

### 5.1 DATASETS

We perform experiments on two language pairs, English-German (high-resource) and English-Japanese (medium-resource).

#### 5.1.1 ENGLISH-GERMAN (EN→DE)

**Base training data** The base model was trained on the en→de WMT'21 training data (Akhbardeh et al., 2021). We used all parallel data, after filtering based on length heuristics (removing source sentences longer than 250 tokens and sentences pairs with a source/target ratio exceeding 1.5) and language id, performing deduplication, and normalizing punctuation. We also used the WMT'21 monolingual NewsCrawl data for backtranslation (with a WMT de→en backtranslation model). After filtering, the training dataset had 89.4 million sentences.

**Finetuning data** We used previous WMT test sets from the years 2009-2019 as finetuning data (Akhbardeh et al., 2021). Our finetuning dataset had a total of 30,426 sentences. The target side of this dataset is human-translated references, and we finetuned both on these targets, as well as on MBR-decoded and QE-reranked translations generated from the source-side sentences.

#### 5.1.2 ENGLISH-JAPANESE (EN→JA)

**Base training data** The base model was trained on the en→ja WMT'22 parallel training data (Kocmi et al., 2022). The data was deduplicated and filtered using CDS (Wang et al., 2018), where the top 30% of data by CDS score was preserved. After filtering, the training dataset had 8 million sentences.

**Finetuning data** We used the WMT'20 test set, comprised of 1000 sentences, as finetuning data (Kocmi et al., 2022). As with en→de, we finetuned both on the provided references, and on MBR and QE translations generated from this dataset's source sentences.

### 5.1.3 MONOLINGUAL DATA

The finetuning datasets for en→de and en→ja only have 30k and 1k examples, respectively, as these datasets are constrained by the requirement of high-quality human reference translations. Since MBR and QE finetuning do not require references, we also experiment with using monolingual datasources. For that, we take a random sample of 200k English sentences from Common Crawl.[1]

### 5.1.4 MBR AND QE DATA

We generate MBR and QE translations both from the en→de and en→ja finetuning datasets (§5.1.1 and §5.1.2) and from the monolingual CommonCrawl dataset (§5.1.3). We generate 256 candidate translations per source via epsilon sampling (setting $\epsilon$=0.02), and use the same set of sampling translations to generate both the MBR and QE data. We use *BLEURT* (Sellam et al., 2020) as the MBR utility metric and *MetricX-XXL-QE*, a QE version of *MetricX-XXL* (Freitag et al., 2022b) with the same architecture and trained on the same human judgements data, as the QE utility metric. In addition to generating MBR and QE translations as finetuning data, we also generate beam search translations from the same datasets as baselines. We use beam size of 4 and length penalty as described in Equation 10 in Wu et al. (2016) with $\alpha = 0.5$.

### 5.1.5 DEVELOPMENT AND TEST SETS

For both language pairs, we use newstest2021 as our development set for checkpoint picking, and report all results on the generalMT2022 test set (Kocmi et al., 2022).

## 5.2 MODELS AND TRAINING RECIPE

**Student model** For both language pairs (en→de and en→ja), we used a $375.4$ million parameter Transformer encoder-decoder architecture, implemented in *lingvo* (Shen et al., 2019). The model architecture is similar to the *transformer-big* setting in Vaswani et al. (2017), with 6 encoder and 6 decoder layers, model dimension of 1024, hidden dimension of 8192, and 16 multi-attention heads. The en→de model used a bilingual vocabulary of 32k subword units and the en→ja model used a multilingual vocabulary of 256k subword units (Kudo & Richardson, 2018). The models were trained without label smoothing to avoid distorting the probability distributions when sampling translations for MBR and QE data generation, as it has been found that label smoothing negatively impacts model fit (Eikema & Aziz, 2021). The best (base and incremental) checkpoints were chosen to maximize *BLEURT* on the development set.

**Teacher models** In addition to self-training, we also experimented with using a LLM as the teacher model. Instead of using a (zero-shot or few-shot) prompt-based approach, we finetuned *PaLM-2 Bison* (Anil et al., 2023) on the (reference-based) finetuning datasets (see §5.1.1 and §5.1.2) to maximize translation quality.

## 5.3 EVALUATION

We evaluate our models on six automatic metrics: *BLEURT* (Sellam et al., 2020), *MetricX-XXL* (Freitag et al., 2022b), *MetricX-23-c* (which resembles *MetricX-XXL* but uses a *PaLM-2* backbone; Juraska et al. (2023)), *chrF* (Popović, 2015), *Comet20* (Rei et al., 2020), and *Comet22* (Rei et al., 2022). Since the MBR data is generated using *BLEURT* as the utility function, and the QE data is generated using *MetricX-XXL-QE*, the MBR-decoded and MBR-finetuned models may overfit to the *BLEURT* metric, while the QE-reranked and QE-finetuned models may overfit to the *MetricX-XXL* metric. So while these metrics can provide useful signals about the effectiveness of the distillation (e.g. from the MBR-decoded teacher to the MBR-finetuned student), they cannot be trusted as unbiased measures of model quality.

---

[1]https://commoncrawl.org

To measure model quality, we instead rely on *Comet20*. Note that *BLEURT* and *Comet20* were finetuned on the same human judgements data, as were *MetricX-XXL* and *MetricX-23-c*, so these pairs of metrics may tend to be highly correlated with each other. Thus, we also report *MetricX-23-c* as a counterbalance to any *Comet20*-specific biases, which may tend to favor models optimized for *BLEURT* (e.g. MBR-decoded and MBR-finetuned models). We also verify the trustworthiness of *Comet20* with a human evaluation study (see Section 5.3.1). Unless otherwise indicated, all evaluation results for the finetuned models are reported using beam search as the decoding strategy (with the same hyperparameters as in §5.1.4).

### 5.3.1 HUMAN EVALUATION

We hired 9 professional translators and measured translation quality with a document context version of MQM (Lommel et al., 2014) which mimics the setup proposed in Freitag et al. (2021). This includes using the same error categories, severity levels, and error weighting schema. We assign a weight of 5 to each major error and 1 to each minor error, except for minor punctuation errors which receive a score of 0.1. The final segment-level score is an average over scores from all annotators. We refer the reader to Freitag et al. (2021) for the details on error categories and annotator instructions.

### 5.4 EXPERIMENT ARMS

We performed experiments in three phases. See Table 6 (Appendix B) for a summary of the datasets used per phase, and see Table 7 for the time and compute costs of generating the MBR and QE data.

**Phase 1** Finetune from the base checkpoint using the finetuning datasets described in §5.1.1 (for en→de) and §5.1.2 (for en→ja). This phase allows for a **direct comparison of MBR and QE finetuning against finetuning on human-generated references**. In the remaining phases, large corpora of unpaired monolingual corpora are used to generate the MBR and QE datasets, so no comparison against reference-based finetuning is possible.

**Phase 2** Finetune from both base and incremental (reference-finetuned) checkpoints using self-MBR and self-QE data generated from the monolingual Common Crawl corpus described in §5.1.3. In this phase, we **investigate whether using a larger corpus of source-side data to generate MBR and QE translations yields additional quality improvements** over the finetuned models from Phase 1, and the extent to which this closes the performance gap with finetuning on references.

**Phase 3** Finetune from both base and incremental (reference-finetuned) checkpoints using MBR and QE data generated from the *PaLM-2 Bison* teacher model. In this phase, we **investigate whether using a teacher model which is stronger than the student affords further quality gains**.

## 6 RESULTS

### 6.1 EN→DE

The *Comet20* results from all experiment phases are summarized in Table 2. We observe that QE reranking (1b) and MBR decoding (1c) using the base checkpoint both outperform beam search decoding (1a). This establishes the premise that the MBR and QE "teachers" used to generate the MBR and QE finetuning data are stronger than the beam search baseline. Also see Table 8 (Appendix C.1) for results across all automatic metrics (as described in §5.3), see Table 9 for a breakout of the *Comet20* results by domain, and see Table 10 for the performance of the *PaLM-2* teacher.

| Model | Comet20 ↑ | MQM score ↓ |
|---|---|---|
| 1a) Beam search decoding (base model) | 57.79 | 1.422 |
| 1b) QE reranking (base model) | 59.21 | 1.289 |
| 1g) Reference-finetuned | 60.95 | 1.183 |
| 2d) Self-QE-from-ref-finetuned | 61.11 | 1.099 |
| 3c) PaLM-2-QE-from-ref-finetuned | 62.35 | 1.085 |

Table 1: Results of MQM study on WMT'22 test set. Lower MQM scores are better (indicating fewer errors).

| Model | COMET20 |
|---|---|
| 0a) WMT 2022 Top Submission (JDExploreAcademy) | 63.26* |
| 1a) Beam search decoding (base model) | 57.79 |
| 1b) QE reranking (base model) | 59.21* (+1.42) |
| 1c) MBR decoding (base model) | 59.35* (+1.56) |
| 1d) Beam-finetuned | 57.08  (-0.71) |
| 1e) MBR-finetuned | 58.02  (+0.23) |
| 1f) QE-finetuned | 59.63* (+1.84) |
| 1g) Reference-finetuned | 60.95* (+3.16) |
| 2a) Self-MBR-from-base-finetuned | 59.94* (+2.15) |
| 2b) Self-beam-from-ref-finetuned | 60.06  (-0.89) |
| 2c) Self-QE-from-base-finetuned | 60.49* (+2.70) |
| 2d) Self-QE-from-ref-finetuned | 61.11  (+0.16) |
| 2e) Self-MBR-from-ref-finetuned | 61.49  (+0.54) |
| 3a) PaLM-2-greedy-from-ref-finetuned | 61.61  (+0.66) |
| 3b) PaLM-2-QE-from-base-finetuned | 62.34* (+4.55) |
| 3c) PaLM-2-QE-from-ref-finetuned | 62.35* (+1.40) |
| 3d) PaLM-2-MBR-from-base-finetuned | 63.51* (+5.72) |
| 3e) PaLM-2-MBR-from-ref-finetuned | 63.62* (+2.67) |

Table 2: Results on en→de WMT'22 test set. Asterisks indicate scores which are significantly better than their baseline and parentheticals indicate delta from baseline. For models finetuned from the base checkpoint, their baseline is row 1a) and for models finetuned from the reference-finetuned checkpoint, their baseline is row 1g).

**Phase 1**  Finetuning on references (1g) achieves the highest *Comet20* score (with a gain of 3.16 points over the base model), though MBR and QE finetuning (1e and 1f, respectively) still achieve gains of 0.23 and 1.84 points against the base model, respectively, in contrast to beam finetuning (1d) which degrades performance.

**Phase 2**  MBR and QE finetuning from the base checkpoint on the larger Common Crawl dataset (2a and 2c, respectively) both yield gains against the base model, and outperform the Phase 1 finetuning on the smaller dataset (1e and 1f, respectively). Moreover, performing a second round of finetuning from the reference-finetuned checkpoint on either self-QE-Common-Crawl or self-MBR-Common-Crawl data (2d and 2e, respectively) achieves further gains in *Comet20*. So MBR and QE finetuning from a large monolingual corpus can complement finetuning on a small dataset of high-quality human references. Also note that beam finetuning (using the same source-side Common Crawl dataset) degrades performance relative to the initial reference-finetuned checkpoint.

**Phase 3**  Using QE and MBR finetuning data generated by the *PaLM-2 Bison* teacher (3b and 3d, respectively) outperforms using the self-teacher model (2c and 2a, respectively) by a large margin (of 1.85 and 3.57 points, respectively) and, in fact, outperforms finetuning on references (by 1.39 and 2.56 points, respectively). Recall that the *PaLM-2 Bison* teacher was finetuned on references, but the student model never directly saw them. Moreover, performing a second round of QE or MBR finetuning from the reference-finetuned checkpoint using the *PaLM-2 Bison* teacher (3c and 3e, respectively) yields further performance improvements, with the MBR-finetuned model achieving an additional large gain of 2.67 points on top of the reference-finetuned model, outperforming all other models including the winning WMT'22 submission.

**Human Evaluation: MQM Study**  We perform a human evaluation to compare the "from-ref" QE-finetuned systems 2d) and 3c) in Table 2 (using a self-teacher and *PaLM-2* teacher, respectively), against the beam search decoded and QE-reranked base model (1a and 1b, respectively), as well as the reference-finetuned model (1g) from which QE finetuning was initialized. As shown in Table 1, the ranking of the systems according to human evaluation agrees with the ranking according to *Comet20*. Both QE-finetuned systems outperform the reference-finetuned baseline, and QE finetuning using the *PaLM-2* teacher outperforms using the self-teacher.

## 6.2   EN→JA

We then investigate whether the gains from QE finetuning also extend to a lower-resource setting. Recall that the human reference-based finetuning dataset for English-Japanese only has 1k sentences,

while monolingual English data is widely available (to use for generating Japanese MBR and QE translations). The results for all en→ja experiment phases are summarized in Table 3, and the trends align with those observed for en→de.

| Model | COMET20 |
|---|---|
| 0a) WMT 2022 Top Submission (NT5) | 64.15* |
| 1a) Beam search decoding (base model) | 47.04 |
| 1b) Reference-finetuned | 50.96* (+3.92) |
| 1c) QE-finetuned | 51.39* (+4.35) |
| 1d) Beam-finetuned | 47.04 (+0.00) |
| 2a) Self-QE-from-base-finetuned | 53.73* (+6.69) |
| 2b) Self-QE-from-ref-finetuned | 56.69* (+5.73) |
| 3a) PaLM-2-QE-from-base-finetuned | 60.63* (+13.59) |
| 3b) PaLM-2-QE-from-ref-finetuned | 60.58* (+9.62) |

Table 3: Results on en→ja WMT'22 test set. Asterisks indicate scores which are significantly better than their baseline and parentheticals indicate delta from baseline. For models finetuned from the base checkpoint, their baseline is row 1a), and for models finetuned from the reference-finetuned checkpoint, their baseline is row 1b).

**Phase 1** QE finetuning (1c) achieves a $4.35$ *Comet20* performance improvement over the base model (1a) and, even more notably, outperforms finetuning on human references (1b) by $0.43$ points. The baseline of beam finetuning (1d), on the other hand, degrades performance relative to the base model. Note that for beam finetuning, the max-*BLEURT* checkpoint on the development set was the initial checkpoint, so the performance reported in row 1d) matches that of the base model (1a), as per our checkpoint selection procedure.

**Phase 2** QE finetuning from the base checkpoint (2a) on the larger Common Crawl dataset (200k sentences, versus 1k from Phase 1) achieves even stronger performance (better by $2.34$ points), outperforming the reference-finetuned model (1b) by $2.77$ points. Moreover, QE finetuning from the reference-finetuned checkpoint (2b) achieves additional large performance improvements, beating the reference-finetuned model by $5.73$ points.

**Phase 3** As with en→de, the largest performance improvements are observed when using the *PaLM-2 Bison* teacher, rather than the self-teacher setup from the preceding phases. Here, we see that QE finetuning from the base checkpoint using the *PaLM-2 Bison* teacher (3a) dramatically outperforms finetuning on references (1b) by $9.67$ points, while QE finetuning from the reference-finetuned checkpoint (3b) does not afford any additional performance improvements.

### 6.3 ABLATION STUDY: WHAT IS THE EFFECT OF THE DECODING STRATEGY ON THE PERFORMANCE OF MBR-FINETUNED AND QE-FINETUNED MODELS?

We present a single ablation study here, and refer the reader to Appendix D for a comprehensive battery of ablation studies covering the effects of the type and size of the (source-side) monolingual dataset and of candidate size used for MBR and QE data generation on the performance of the finetuned model, as well as the performance implications of finetuning on back-translated MBR and QE data, mixtures of MBR and QE data, and iterative finetuning.

All MBR-finetuned and QE-finetuned model results reported so far use beam search decoding. The techniques of MBR and QE finetuning rely on using an efficient decoding algorithm, but this does introduce a mismatch between the decoding algorithm used to generate the finetuning data and the decoding algorithm used by the student at inference time. Greedy search and sampling are also efficient decoding algorithms, so we compare their performance against that of beam search, using the best en→de MBR-finetuned and QE-finetuned models from each experiment phase. We use epsilon sampling with $\epsilon = 0.02$ as the sampling strategy. As shown in Table 4, beam search outperforms the other decoding methods for the Phase 1 models, while the even-more-efficient greedy decoding is a close contender and actually outperforms beam search for the Phase 2 and Phase 3 QE-finetuned models. Sampling decoding, on the other hand, lags well behind beam search and greedy decoding. Also see Table 20 in Appendix D for a comparison of how MBR decoding and QE reranking compare against these more efficient decoding strategies for MBR-finetuned and QE-finetuned models.

|              | Model    | QE-finetuned | MBR-finetuned |
|--------------|----------|--------------|---------------|
| *Phase 1*    | Beam     | 59.63        | 58.02         |
|              | Greedy   | 59.58        | 57.61         |
|              | Sampling | 55.91        | 57.69         |
| *Phase 2* (from-ref) | Beam     | 61.11        | 61.49         |
|              | Greedy   | 61.19        | 60.93         |
|              | Sampling | 57.14        | 57.80         |
| *Phase 3* (from-ref) | Beam     | 62.35        | 63.62         |
|              | Greedy   | 62.52        | 62.65         |
|              | Sampling | 56.22        | 58.92         |

Table 4: Performance of en→de QE and MBR-finetuned models when using different decoding strategies.

# 7 DISCUSSION

High-quality datasets of human-translated references are expensive to curate. We have seen that, for both a high-resource (en→de) and medium-resource (en→ja) language pair, MBR and QE finetuning lead to significant improvements over the base model. Moreover, self-MBR and self-QE finetuning on top of the reference-finetuned model (already a strong baseline) further boost performance, while performance improvements are not observed when finetuning on self-beam translations. We also showed that using an external LLM teacher to generate MBR or QE finetuning data leads to even larger performance improvements and, in fact, outperforms finetuning on human-translated references.

Given the effectiveness of an external teacher model (which differs in architecture, training data, etc. from the student) relative to a self-teacher, the success of these finetuning methods can likely *not* be primarily explained by mitigation of the label exposure bias problem (Schmidt, 2019). Instead, the effectiveness of MBR and QE finetuning seemingly can be attributed primarily to the high quality of the translations used as finetuning data.

When comparing MBR against QE finetuning, we see that QE tends to perform at least as well as MBR finetuning using a self-teacher, while MBR outperforms QE finetuning using the LLM teacher. We hypothesize that the sampling translations from the self-teacher do not closely approximate the true reference distribution, so using them as pseudo-references does not help (or even hurts), relative to using a reference-free (QE) metric. For higher-quality models (e.g. *PaLM-2*), the candidate translations are good enough approximations to references that they provide a useful signal.

MBR and QE finetuning shift the MBR and QE quality gains to training time, with the expectation that an efficient decoding strategy be used at inference time. When evaluating the finetuned models, we showed that beam search and greedy decoding perform comparably, and outperform sampling. Note that these decoding methods incrementally continue a partial translation hypothesis, while MBR decoding and QE reranking have the advantage that they operate on full translation hypotheses. Given that MBR and QE finetuning improve performance even using efficient decoding methods, these finetuning techniques are indeed able to help the model make better local decisions during decoding.

# 8 CONCLUSION

In this work, we have proposed the techniques of *MBR finetuning* and *QE finetuning* to distill the quality gains from MBR decoding and QE reranking, which are state-of-the-art decoding methods for NLG models, but are prohibitively expensive to compute. We have shown that MBR and QE finetuning lead to consistent quality improvements over the base model, even when using a self-teacher, and also achieve additional performance improvements on top of human reference-finetuned models. Furthermore, using a LLM as the teacher model substantially outperforms using a self-teacher and, in fact, MBR and QE finetuning using a LLM teacher outperforms finetuning on references. For most language pairs, (source-side) monolingual data is available in larger quantities than parallel data. MBR and QE finetuning open up the possibility of leveraging monolingual data to achieve substantial gains in model performance both through self-training and distillation from a stronger teacher model.

There are many avenues for future work, including extending MBR and QE finetuning to other NLG tasks besides NMT, more comprehensively investigating the effect of the utility function used on MBR and QE finetuning performance, experimenting with few-shot generation of MBR and QE data from a LLM teacher (removing the need for finetuning the LLM), and document-level MBR and QE finetuning.

## 9 REPRODUCIBILITY STATEMENT

This work was performed on open-source WMT datasets (Akhbardeh et al., 2021; Kocmi et al., 2022) using the open-source *lingvo* library (Shen et al., 2019).

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

## A  MBR AND QE UTILITY FUNCTIONS

| Metric | Seg-Level acc$^\star$ | Sys-Level acc |
|---|---|---|
| BLEURT | 57.5 | 76.9 |
| METRICX-XXL-QE | 59.1 | 76.9 |

Table 5: Segment-level and system-level meta-evaluation results on the WMT'22 MQM en→ de test set.

Table 5 shows the meta-evaluation of the (reference-based) *BLEURT* metric against the (reference-free) *MetricX-XXL-QE* metric. We measure the metric performance with respect to ground-truth translation quality ratings using the benchmark WMT'22 en→de MQM dataset. At the segment-level, we report the "group-by-item" variant of the pairwise accuracy meta-evaluation metric proposed by Deutsch et al. (2023). Note that a random metric would achieve 33% accuracy. We also report system-level pairwise accuracy. According to both the segment-level and system-level meta-evaluation metrics, *BLEURT* and *MetricX-XXL-QE* perform comparably.

Figures 1 and 2 present additional comparisons of the scores generated via MBR decoding versus QE reranking.

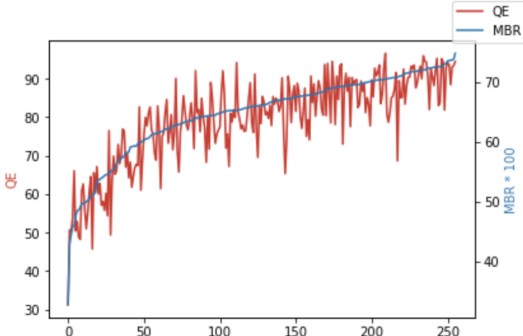

Figure 1: QE vs MBR scores for all 256 candidate translations generated from a single source sentence in the en→de newstest2009-2019 dataset. The ranking of examples by QE score is not the same as the ranking by MBR score.

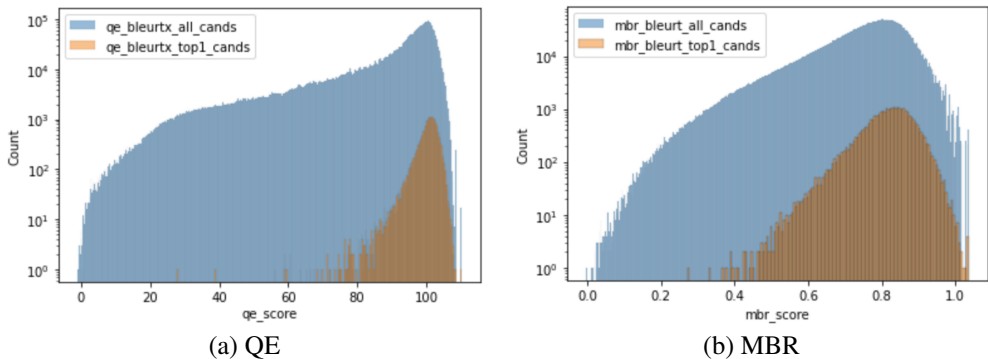

Figure 2: QE and MBR score distributions for all candidates versus the top-1 candidate (for candidate_size=256), generated from the (source-side) en→de NewsTest 2009-2019 finetuning set.

## B  MBR AND QE DATASET STATISTICS

See Table 6 for a summary of the (source-side) datasets used to generate MBR and QE data during each of the three experimental phases. Note that during Phase 1 and Phase 2, the "self-teacher" was used, and during Phase 3, the *PaLM-2 Bison* teacher was used.

| | Base training | Phase 1 | Phase 2 | Phase 3 |
|---|---|---|---|---|
| **en→de** | *WMT 2021 parallel training data (filtered)* 89.4 million | *WMT 2009-2019 (newstest) test sets* 30,426 | *Common Crawl* 200,000 | *Common Crawl* 200,000 |
| **en→ja** | *WMT 2022 parallel training data (filtered)* 8 million | *WMT 2020 test set* 1,000 | *Common Crawl* 200,000 | *Common Crawl* 200,000 |

Table 6: Training and finetuning datasets used, and their respective sizes, at each experimental phase.

The bottleneck in generating data for MBR and QE finetuning is in the candidate scoring stage. Table 7 shows the time and compute requirements for performing MBR-BLEURT and QE-MetricX-XXL scoring, across all three experiment phases and across both language pairs. We use TPUs for BLEURT and MetricX-XXL scoring, as described in Jouppi et al. (2020). The TPUv4 hardware is a faster and newer generation than TPUv3. MBR scoring takes about 7.65 TPUv4 minutes/source sentence, while QE scoring takes about 4.2 TPUv3 seconds/source sentence. So even when using faster hardware for MBR, QE scoring is 109x faster.

| en→de | | Phase 1 | Phase 2 | Phase 3 |
|---|---|---|---|---|
| | | *WMT 2009-2019 test sets* | *Common Crawl* | *Common Crawl* |
| | **MBR** | 7.8 hours (500 TPUv4) | 5.1 hours (5000 TPUv4) | 5.1 hours (5000 TPUv4) |
| | **QE** | 1.1 hours (32 TPUv3) | 7.3 hours (32 TPUv3) | 7.3 hours (32 TPUv3) |
| **en→ja** | | *WMT 2020 test set* | *Common Crawl* | *Common Crawl* |
| | **QE** | 2.2 minutes (32 TPUv3) | 7.3 hours (32 TPUv3) | 7.3 hours (32 TPUv3) |

Table 7: Wallclock time for scoring step during MBR and QE data generation. TPUv4 is a newer and faster generation than TPUv3 (Jouppi et al., 2020).

## C    ADDITIONAL SYSTEM COMPARISON

### C.1    EN→DE RESULTS

Table 8 shows en→de results across all metrics (while the main text only reports *Comet20*).

| Model | BLEURT | MetricX-XXL | MetricX-23-c | chrF | COMET22 | COMET20 |
|---|---|---|---|---|---|---|
| 0a) WMT 2022 Top Submission (JDExploreAcademy) | 78.38 | 82.71 | -0.61 | 86.04 | 87.41 | 63.26* |
| 1a) Beam search decoding (base model) | 76.78 | 80.52 | -0.73 | 82.54 | 85.84 | 57.79 |
| 1b) QE reranking (base model) | 78.01 | 82.92 | -0.55 | 82.83 | 86.56 | 59.21* |
| 1c) MBR decoding (base model) | 81.09 | 82.29 | -0.67 | 82.73 | 86.38 | 59.35* |
| 1d) Beam-finetuned | 76.14 | 79.99 | -0.77 | 82.76 | 85.65 | 57.08 |
| 1e) MBR-finetuned | 77.34 | 80.36 | -0.75 | 83.46 | 85.83 | 58.02 |
| 1f) QE-finetuned | 77.24 | 81.25 | -0.70 | 83.33 | 86.34 | 59.63* |
| 1g) Reference-finetuned | 77.49 | 81.53 | -0.68 | 83.51 | 86.64 | 60.95* |
| 2a) Self-MBR-from-base-finetuned | 77.79 | 81.42 | -0.69 | 83.51 | 86.25 | 59.94* |
| 2b) Self-beam-from-ref-finetuned | 77.00 | 81.01 | -0.70 | 83.27 | 86.34 | 60.06 |
| 2c) Self-QE-from-base-finetuned | 77.57 | 81.43 | -0.70 | 84.27 | 86.10 | 60.49* |
| 2d) Self-QE-from-ref-finetuned | 77.86 | 81.63 | -0.67 | 83.53 | 86.75 | 61.11 |
| 2e) Self-MBR-from-ref-finetuned | 78.50 | 81.72 | -0.66 | 83.45 | 86.76 | 61.49 |
| 3a) PaLM-2-greedy-from-ref-finetuned | 77.60 | 81.62 | -0.69 | 83.46 | 86.73 | 61.61 |
| 3b) PaLM-2-QE-from-base-finetuned | 78.21 | 82.22 | -0.66 | 84.14 | 86.85 | 62.34* |
| 3c) PaLM-2-QE-from-ref-finetuned | 78.26 | 82.15 | -0.65 | 84.10 | 87.11 | 62.35* |
| 3d) PaLM-2-MBR-from-base-finetuned | 79.35 | 82.82 | -0.64 | 84.55 | 86.96 | 63.51* |
| 3e) PaLM-2-MBR-from-ref-finetuned | 79.39 | 82.85 | -0.63 | 84.51 | 87.29 | 63.62* |

Table 8: Results on English-German WMT'22 test set across all metrics. Asterisks in the *Comet20* column indicate scores which are significantly better than their baseline. For models finetuned from the base checkpoint, their baseline is row 1a), and for models finetuned from the reference-finetuned checkpoint, their baseline is row 1g).

Table 9 shows the en→de *Comet20* results broken out by domain (Conversation, e-Commerce, News, and Social).

| Model | Conversation | e-Commerce | News | Social |
|---|---|---|---|---|
| 0a) WMT 2022 Top Submission (JDExploreAcademy) | 69.40 | 66.64 | 63.78 | 53.12 |
| 1a) Beam search decoding (base model) | 63.49 | 62.51 | 59.61 | 45.69 |
| 1b) QE reranking (base model) | 62.57 | 64.17 | 59.58 | 50.39 |
| 1c) MBR decoding (base model) | 64.90 | 62.98 | 60.88 | 48.53 |
| 1d) Beam-finetuned | 62.27 | 62.45 | 58.92 | 44.32 |
| 1e) MBR-finetuned | 62.68 | 62.04 | 60.74 | 46.58 |
| 1f) QE-finetuned | 62.88 | 64.60 | 60.94 | 49.63 |
| 1g) Reference-finetuned | 66.23 | 65.07 | 62.63 | 49.74 |
| 2a) Self-MBR-from-base-finetuned | 66.53 | 62.93 | 60.91 | 49.30 |
| 2b) Self-beam-from-ref-finetuned | 66.01 | 64.31 | 61.37 | 48.40 |
| 2c) Self-QE-from-base-finetuned | 66.37 | 63.99 | 62.17 | 49.65 |
| 2d) Self-QE-from-ref-finetuned | 66.61 | 64.87 | 62.27 | 50.55 |
| 2e) Self-MBR-from-ref-finetuned | 68.42 | 64.76 | 61.53 | 51.16 |
| 3a) PaLM-2-greedy-from-ref-finetuned | 68.11 | 65.39 | 62.24 | 50.63 |
| 3b) PaLM-2-QE-from-base-finetuned | 67.08 | 66.43 | 63.51 | 52.46 |
| 3c) PaLM-2-QE-from-ref-finetuned | 68.07 | 65.60 | 63.44 | 52.23 |
| 3d) PaLM-2-MBR-from-base-finetuned | 70.04 | 67.11 | 64.11 | 53.00 |
| 3e) PaLM-2-MBR-from-ref-finetuned | 71.10 | 66.44 | 63.96 | 52.93 |

Table 9: *Comet20* scores on English-German WMT'22 test set, broken out by domain.

Table 10 shows the performance of the en→de *PaLM-2* greedy-decoded, QE-reranked, and MBR-decoded teacher models.

| Model | BLEURT | MetricX-XXL | COMET20 |
|---|---|---|---|
| PaLM-2 greedy decoding | 78.81 | 83.59 | 63.23 |
| PaLM-2 QE reranking | 78.77 | 84.29 | 60.86 |
| PaLM-2 MBR decoding | 82.43 | 84.73 | 63.76 |

Table 10: Performance of greedy-decoded, QE-reranked, and MBR-decoded en→de *PaLM-2* teacher models on the WMT'22 test set.

Figure 3 shows a comparison of the similarity of the Phase 1 en→de self-MBR and self-QE finetuned models against their teachers, as measured by cross-BLEU on the WMT'22 test set. Note that the similarity between each teacher and respective student is lower than the similarity between the finetuned models.

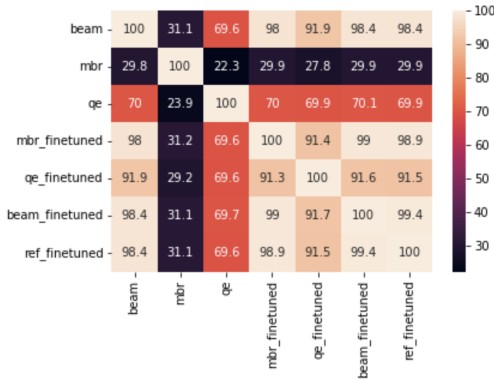

Figure 3: Cross-BLEU of en→de models on WMT'22 test set, as a measure of model similarity.

## C.2 EN→JA RESULTS

Table 11 shows en→ja results across all metrics (while the main text only reports *Comet20*).

| Model | BLEURT | MetricX-XXL | MetricX-23-c | chrF | COMET22 | COMET20 |
|---|---|---|---|---|---|---|
| 0a) WMT 2022 Top Submission (NT5) | 69.49 | 83.24 | -0.62 | 95.15 | 89.26 | 64.15* |
| 1a) Beam search decoding (base model) | 64.69 | 77.28 | -0.95 | 91.19 | 86.03 | 47.04 |
| 1b) Reference-finetuned | 65.70 | 78.13 | -0.92 | 92.01 | 86.81 | 50.96* |
| 1c) QE-finetuned | 65.71 | 78.35 | -0.89 | 94.30 | 86.67 | 51.39* |
| 1d) Beam-finetuned | 64.69 | 77.28 | -0.95 | 91.19 | 86.03 | 47.04 |
| 2a) Self-QE-from-base-finetuned | 65.52 | 78.36 | -0.84 | 94.46 | 87.21 | 53.73* |
| 2b) Self-QE-from-ref-finetuned | 66.56 | 79.31 | -0.81 | 94.32 | 87.76 | 56.69* |
| 3a) PaLM-2-QE-from-base-finetuned | 68.02 | 80.51 | -0.77 | 95.40 | 88.5 | 60.63* |
| 3b) PaLM-2-QE-from-ref-finetuned | 67.94 | 80.35 | -0.78 | 95.30 | 88.48 | 60.58* |

Table 11: Results on English-Japanese WMT'22 test set across all metrics. Asterisks in the *Comet20* column indicate scores which are significantly better than their baseline. For models finetuned from the base checkpoint, their baseline is row 1a), and for models finetuned from the reference-finetuned checkpoint, their baseline is row 1b).

Table 12 shows the performance of the en→ja *PaLM-2* greedy-decoded, QE-reranked, and MBR-decoded teacher models.

| Model | BLEURT | MetricX-XXL | COMET20 |
|---|---|---|---|
| PaLM-2 greedy decoding | 71.51 | 85.56 | 70.25 |
| PaLM-2 QE reranking | 70.17 | 86.37 | 67.52 |
| PaLM-2 MBR decoding | 75.64 | 86.44 | 72.69 |

Table 12: Performance of en→ja greedy-decoded, QE-reranked, and MBR-decoded *PaLM-2* teacher models on the WMT'22 test set.

### C.3  DE→EN RESULTS

We show Phase 1 results for German-English in Table 13. We perform MBR finetuning and QE finetuning using the (source side of the) WMT 2009-2019 test sets (30,695 examples total; Akhbardeh et al. (2021)), and compare this against finetuning on the references and on beam translations. We use the same model architecture as for en→de and en→ja, described in §5.2.

| Model | BLEURT | MetricX-XXL | chrF | COMET22 | COMET20 |
|---|---|---|---|---|---|
| 1a) Base model | 72.96 | 78.03 | 76.55 | 84.17 | 51.77 |
| 1b) Beam-finetuned | 72.97 | 78.02 | 75.94 | 84.18 | 51.77 |
| 1c) MBR-finetuned | 73.62 | 78.45 | 76.74 | 84.38 | 53.31 |
| 1d) Reference-finetuned | 73.64 | 78.76 | 76.00 | 84.67 | 54.15 |
| 1e) QE-finetuned | 73.86 | 78.99 | 76.54 | 84.81 | 55.10 |

Table 13: Results on de→en WMT'22 test set. Beam search is used as the decoding strategy for all models.

We find that the trends for this (into-English) language pair align with those observed for the other (out-of-English) language pairs in our study. In particular, we observe that MBR finetuning and QE finetuning both improve performance relative to the base model, while beam finetuning does not. Moreover, QE finetuning outperforms finetuning on references (as also observed for en→ja).

## D  ADDITIONAL ABLATION STUDIES

There are many source-side and target-side variables which may affect the performance of MBR and QE finetuning. We perform ablation studies (on en→de) to isolate the effects of several of these variables. On the source side, we consider the effect of the type and size of monolingual dataset used for MBR and QE data generation. On the target side, we investigate the effect of candidate size used for MBR and QE data generation on the performance of the finetuned model. We also explore finetuning on back-translated MBR and QE data, mixtures of MBR and QE data, as well as iterative finetuning.

### D.0.1  DOES THE MONOLINGUAL DATASET USED FOR MBR AND QE FINETUNING MATTER?

To probe the effect of the source-side dataset on the downstream performance of MBR and QE finetuning, we sample 30k sentences from the NewsCrawl (Akhbardeh et al., 2021) and Common-Crawl (§5.1.3) datasets, then use the en→de reference-finetuned checkpoint to generate MBR and QE translations, which are used for a second round of finetuning. As shown in Table 14, choice of dataset matters very little. The NewsCrawl dataset has a small advantage over CommonCrawl in *Comet20* scores for the MBR-finetuned model, while the reverse is true for the QE-finetuned model.

| Model | BLEURT | MetricX-XXL | COMET20 |
|---|---|---|---|
| MBR-finetuned (NewsCrawl) | 78.11 | 81.85 | 60.60 |
| MBR-finetuned (CommonCrawl) | 78.33 | 81.72 | 60.06 |
| QE-finetuned (NewsCrawl) | 77.64 | 81.57 | 59.40 |
| QE-finetuned (CommonCrawl) | 77.70 | 81.53 | 60.62 |

Table 14: Comparison of finetuning results using 30k sentences sampled from NewsCrawl versus CommonCrawl datasets.

We also investigate the extent to which dataset size affects model performance, by finetuning on the entire Common Crawl (§5.1.3) MBR and QE dataset (generated from the reference-finetuned checkpoint) of 200k sentences, versus finetuning on a sample of 30k sentences. The results are shown in Table 15. Reducing the dataset to only 15% of its original size leads to a consistent, but small, decrease in performance across all metrics.

| Model | BLEURT | MetricX-XXL | COMET20 |
|---|---|---|---|
| MBR-finetuned (CC-200k) | 78.50 | 81.72 | 61.49 |
| MBR-finetuned (CC-30k) | 78.33 | 81.72 | 60.06 |
| QE-finetuned (CC-200k) | 77.86 | 81.63 | 61.11 |
| QE-finetuned (CC-30k) | 77.70 | 81.53 | 60.62 |

Table 15: Comparison of MBR and QE finetuning (from the reference-finetuned checkpoint) using Common-Crawl dataset samples of size 200k vs 30k.

### D.1 HOW DOES THE NUMBER OF CANDIDATE TRANSLATIONS USED FOR QE-RERANKED DATASET GENERATION AFFECT DOWNSTREAM QE FINETUNING PERFORMANCE?

As shown in Figure 4, finetuning on a QE dataset generated with candidate_size=256 does *not* outperform candidate_size=32. For QE-finetuned models across all candidate sizes (32, 64, 128, and 256), their performance exceeds not only the (beam search decoded) checkpoint from which finetuning is initialized, but also the performance of their QE-reranked teacher model. So even with a small candidate size (allowing for efficient dataset generation), the performance of the teacher model can be exceeded.

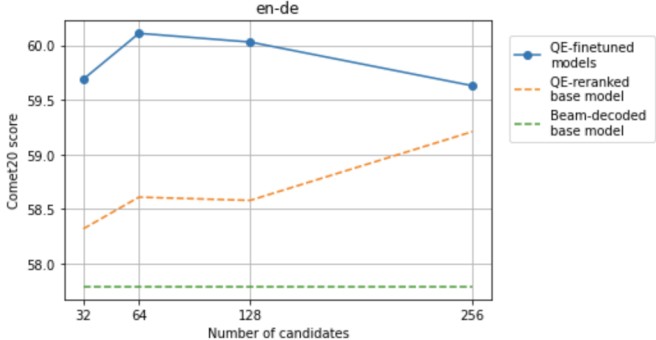

Figure 4: Finetuned model performance as a function of the number of candidates per source used to generate the QE dataset. Performance of the QE-reranked teacher model improves as the candidate size increases, while the QE-finetuned student is more robust to candidate size, and outperforms the teacher at all candidate sizes. The lower bound of beam search decoding from the base model is also shown for perspective.

### D.2 AT WHAT DATASET SIZE DOES QE FINETUNING ACHIEVE PARITY WITH FINETUNING ON REFERENCES?

In §6.1, we saw that finetuning on references outperforms self-QE finetuning given the same source-side dataset (of 30k sentences). But for many language pairs, high-quality human-translated data suitable for incremental training is scarce. In this ablation, we finetuned on subsamples from the references dataset to determine the reference dataset size at which parity with QE finetuning is achieved.

As shown in Table 16, QE finetuning achieves slightly superior performance (in terms of *Comet20* score) to finetuning on a references dataset of 5k sentences. So for this particular setup (language pair, dataset, model), there is a 6x multiplier in QE dataset size relative to references dataset size needed to achieve performance parity.

| Model | BLEURT | MetricX-XXL | COMET20 |
|---|---|---|---|
| Reference-finetuned (30k) | 77.49 | 81.53 | 60.95 |
| Reference-finetuned (20k) | 77.31 | 81.27 | 60.48 |
| Reference-finetuned (10k) | 77.29 | 81.22 | 60.07 |
| Reference-finetuned (5k) | 77.35 | 81.25 | 59.60 |
| QE-finetuned (30k) | 77.24 | 81.25 | 59.63 |

Table 16: Reference-finetuning on subsampled datasets of different sizes. The *Comet20* score of the reference-finetuned model on 5k sentences is slightly lower than that of the QE-finetuned model on 30k sentences.

### D.3 IS BACK-TRANSLATED QE DATA AS EFFECTIVE AS FORWARD-TRANSLATED?

In §6.1, we showed that QE finetuning outperforms the baseline of finetuning on beam search translations. Finetuning on back-translated QE data is another useful baseline to compare against, to understand whether or not the decoder specifically benefits from learning from the quality and style of (target-side) QE translations.

To generate back-translated QE data, we used a WMT German-English model with the same architecture as described in 5.2. We generated QE translations of the NewsTest 2009-2019 German-English corpus, and used these translations to finetune the base English-German model. As shown in Table 17, finetuning on back-translated QE data doesn't boost model quality as much as finetuning on forward-translated QE data does.

| Model | BLEURT | MetricX-XXL | COMET20 |
|---|---|---|---|
| Base model | 76.78 | 80.52 | 57.79 |
| QE-finetuned | 77.24 | 81.25 | 59.63 |
| QE-BT-finetuned | 76.52 | 80.63 | 59.12 |

Table 17: Comparison of finetuning results using forward-translated vs back-translated QE data.

### D.4 HOW DOES FINETUNING ON A MIXTURE OF MBR AND QE DATA COMPARE TO FINETUNING ON EACH DATASET SEPARATELY?

The mixture was created as a balanced combination of the MBR and QE data generated for the Phase 1 experiments (as described in §18). As shown in Table 18, the *Comet20* score of the mixture-finetuned model falls between the scores of the MBR-finetuned and QE-finetuned models, but is closer to the (higher) score of the QE-finetuned model.

| Model | BLEURT | MetricX-XXL | COMET20 |
|---|---|---|---|
| Base model | 76.78 | 80.52 | 57.79 |
| MBR-finetuned | 77.34 | 80.36 | 58.02 |
| QE-finetuned | 77.24 | 81.25 | 59.63 |
| MBR+QE-finetuned | 77.42 | 81.19 | 59.40 |

Table 18: Comparison of finetuning on MBR data only and QE data only, versus finetuning on a mixture of both datasets.

### D.5 DOES ITERATIVE QE FINETUNING OUTPERFORM A SINGLE ROUND OF FINETUNING?

QE finetuning can be performed iteratively, by generating new QE data for the next finetuning round from the previous round's QE-finetuned checkpoint. We experiment with two rounds of QE finetuning using the setup from Phase 1 (§5.4), with the NewsTest 2009-2019 dataset used for both rounds. We see in Table 19 that a single round of QE finetuning outperforms two rounds.

| Model | BLEURT | MetricX-XXL | COMET20 |
|---|---|---|---|
| QE-finetuned (after 1 round) | 77.24 | 81.25 | 59.63 |
| QE-finetuned (after 2 rounds) | 77.19 | 80.48 | 58.06 |

Table 19: Iterative QE finetuning does not outperform a single round of QE finetuning.

### D.6 DO MBR DECODING AND QE RERANKING IMPROVE THE PERFORMANCE OF MBR-FINETUNED AND QE-FINETUNED MODELS, RELATIVE TO MORE EFFICIENT DECODING METHODS?

As shown in Table 20, after MBR finetuning, QE reranking works better than MBR decoding. Similarly, after QE finetuning, MBR decoding works better than QE reranking. This suggests that using the same decoding method both to generate the finetuning data and at inference time overfits to the metric used as the utility function. Also note that the gap between MBR decoding/QE reranking and beam search/greedy decoding is small for the MBR/QE-finetuned models. In fact, for Phases 2 and 3 of the QE-finetuned models, as well as Phase 3 of the MBR-finetuned model, beam search or greedy decoding is actually the top-performing decoding strategy.

|  | Model | QE-finetuned | MBR-finetuned |
|---|---|---|---|
| *Phase 1* | Beam | 59.63 | 58.02 |
|  | Greedy | 59.58 | 57.61 |
|  | Sampling | 55.91 | 57.69 |
|  | MBR | **60.32** | 55.21 |
|  | QE | 60.13 | **60.03** |
| *Phase 2* (from-ref) | Beam | 61.11 | 61.49 |
|  | Greedy | **61.19** | 60.93 |
|  | Sampling | 57.14 | 57.80 |
|  | MBR | 61.15 | 59.35 |
|  | QE | 60.92 | **61.62** |
| *Phase 3* (from-ref) | Beam | 62.35 | **63.62** |
|  | Greedy | **62.52** | 62.65 |
|  | Sampling | 56.22 | 58.92 |
|  | MBR | 61.76 | 62.30 |
|  | QE | 60.56 | 63.05 |

Table 20: *Comet20* performance of en→de QE-finetuned and MBR-finetuned models when using different decoding strategies, including MBR decoding and QE reranking. Here, the sampling strategy used was epsilon sampling with $\epsilon = 0.02$.

