# OpenReview forum: "MBR and QE Finetuning: Training-time Distillation of the Best and Most Expensive Decoding Methods"
_ICLR.cc/2024/Conference — ICLR 2024 poster_

### Official Review · Reviewer_t5J4 · 2023-11-02

**Soundness:** 3 good
**Presentation:** 3 good
**Contribution:** 2 fair
**Rating:** 6
**Confidence:** 5

**Summary:**

Previous research has shown that ranking techniques (with QE metrics or MBR) generally improve machine translation quality, being very expensive at inference time. This paper proposes MBR/QE finetuning—using QE/MBR outputs at training time, while employing efficient decoding algorithms (e.g., beam search) during inference. Their experiments suggest that (1) MBR/QE finetuning alone are worse than vanilla finetuning with human references using the same high-quality data; (2) self-MBR/QE finetuning can complement finetuning on a small dataset of high-quality human references; and (3) using MBR/QE data generated by a stronger teacher model further improves quality. A subset of the results is validated by professional translators.

**Strengths:**

- The motivation is clear and the paper is well written.
- MBR/QE finetuning (their method) seems to work well with beam/greedy search, making inference faster when compared to standard reranking techniques that employ rerankers at inference time.
- Their models are evaluated with multiple state of the art evaluation metrics for MT. Their findings are further validated by 9 professional translators using the MQM framework.

**Weaknesses:**

- Some parts of the paper (e.g., the first three paragraphs of the introduction) talk about NLG in general but the paper focuses on machine translation. Reranking methods, in particular, rely on good quality estimation models that exist for MT but may not exist for other tasks. Of course, MBR finetuning can be applied to other NLG tasks (e.g., summarization) but this paper does not touch this problem. I think the authors should either revise the writing and make the scope more clear from the beginning, or perform experiments in other NLG tasks. The contribution is mainly empirical and only validated for MT.
- The study is limited in terms of language coverage. They perform experiments on 2 language pairs only (mid & high resource), both for translation out of English. It’s not clear if the findings hold for lower resource languages and when translating into English. In particular, I’m expecting quality estimation models to be worse for low resource languages, which may end up affecting the quality of the translations produced with their method. Have you tried other languages? If so, what happens in that case?
- The related work discusses other methods for training NMT models beyond MLE (e.g., RL methods) but none of them is used as a baseline.

**Questions:**

I left some questions in the weaknesses part. Other comments/questions:

- MBR decoding uses BLEURT and is prohibitively expensive. Note, however, that more efficient alternatives exist. See implementation details discussed in [1], Section 4.3. Is there a reason to choose BLEURT? While efficiency at inference time is an advantage, MBR/QE finetuning is expensive at training time. It would be good to see some numbers to understand the training/inference time difference when compared to other existing approaches.
- In the introduction you say that MBR decoding requires the generation of a large number of candidates. While this is true if you get unbiased samples from the model, it does not seem to be the case when you bias the distribution (e.g., using nucleus sampling). I suggest you see the discussions in [2] and [3] and comment. Also, see my comment above about [1].
- Results use COMET-20 instead of more recent versions already available online such as COMET-22. Is there a reason for using the 2020 version?
- Human evaluation results are important and should not be in the appendix (Table 7). In fact, I think they should be more highlighted in the paper! It would also be interesting to see if they generalize for En-Ja. Can you explain the reasoning for using 9 professional translators for En-De instead of using fewer and evaluate En-Ja as well?
- What happens if you decode with reranking techniques (QE/MBR) using a model trained with your method, instead of using beam/greedy search? Even though this would make the method very inefficient at inference time, it would be interesting to see if it further boosts the performance.
- According to Table 3, both beam and greedy search work well. Does this mean that your method helps solve the beam search curse [4]? What happens when you increase the beam width?

[1] Identifying Weaknesses in Machine Translation Metrics Through Minimum Bayes Risk Decoding: A Case Study for COMET (Amrhein & Sennrich, AACL-IJCNLP 2022)

[2] Quality-Aware Decoding for Neural Machine Translation (Fernandes et al., NAACL 2022).

[3] An Empirical Study of Translation Hypothesis Ensembling with Large Language Models (Farinhas et al., EMNLP 2023).

[4] Six Challenges for Neural Machine Translation (Koehn & Knowles, NGT 2017).

---

> ### Author Response · Authors · 2023-11-17
> **Response to Reviewer t5J4**
>
> Dear Reviewer,
>
> Thank you for your very careful and thoughtful review of our paper. We appreciate your detailed feedback and will seek to individually address each your outstanding concerns here.
>
> 1. *Some parts of the paper (e.g., the first three paragraphs of the introduction) talk about NLG in general but the paper focuses on machine translation. Reranking methods, in particular, rely on good quality estimation models that exist for MT but may not exist for other tasks. Of course, MBR finetuning can be applied to other NLG tasks (e.g., summarization) but this paper does not touch this problem. I think the authors should either revise the writing and make the scope more clear from the beginning, or perform experiments in other NLG tasks. The contribution is mainly empirical and only validated for MT.*
>
> MT is a common testbed for NLG tasks. In the abstract, we state the paper's scope: " Using the canonical NLG task of Neural Machine Translation (NMT), ...". Moreover, while the first three paragraphs of the introduction discuss NLG in general (establishing the motivation and setup in the most general terms), the fourth paragraph already narrows the focus to MT: "In this work, we focus on the NLG task of Neural Machine Translation (NMT)...". In the conclusion, we propose "extending MBR and QE finetuning to other NLG tasks besides NMT" as an avenue for future work.
>
> Note that MBR decoding has been successfully applied to other NLG tasks including text summarization, image captioning, and data-to-text using BERTScore as the utility function, as described in https://arxiv.org/pdf/2311.05263.pdf (Model-based Minimum Bayes Risk Decoding; Jinnai et al., 2023). So applying MBR finetuning to these other tasks would be a natural extension of this work, and would likely perform well.
>
> All of this said, given your concerns about the clarity of the scope, we have revised the first three paragraphs of the introduction, to make explicit from the very beginning that the scope is limited to MT.
>
> 2. *The study is limited in terms of language coverage. They perform experiments on 2 language pairs only (mid & high resource), both for translation out of English. It’s not clear if the findings hold for lower resource languages and when translating into English. In particular, I’m expecting quality estimation models to be worse for low resource languages, which may end up affecting the quality of the translations produced with their method. Have you tried other languages? If so, what happens in that case?*
>
> The quality of MT metrics on low-resource language pairs and into-English language pairs has been evaluated in the WMT Metrics Shared Tasks. See results from WMT'22 (https://aclanthology.org/2022.wmt-1.2.pdf)  and WMT'23 (https://www2.statmt.org/wmt23/pdf/2023.wmt-1.51.pdf), which show that 1) the MetricX metric (including the QE variant) generalizes well to language pairs unseen during training, including low-resource language pairs (e.g. see Hebrew-English in Table 8 of the WMT'23 Metrics Shared Task Report) and 2) BLEURT and MetricX-XXL perform well on into-English language pairs. Note that result 2) is not surprising given that most neural MT metrics, including BLEURT and MetricX-XXL, were finetuned on large language models which were pretrained mostly on English data.
>
> While this paper focuses on mid and high-resource language pairs, existing evidence about the quality of utility functions for low-resource language pairs and into-English language pairs suggests that MBR and QE finetuning would likely work in these settings as well. This would be a fruitful avenue for future work.
>
> 3. *The related work discusses other methods for training NMT models beyond MLE (e.g., RL methods) but none of them is used as a baseline.*
>
> For a correct comparison to RL baselines, we would need to implement them in our framework, to be able to run them on the same data and using the same utility function. Additionally, we would need to tune the RL settings to ensure a fair comparison. Thus, evaluating RL baselines is outside of the scope of this paper. Moreover, the MBR and QE finetuning methods proposed here use MLE, so comparison with RL methods would be tangential to this paper's focus.
>
> (Please see the follow-up comment for individual responses to questions.)

---

> > ### Author Response · Authors · 2023-11-17
> > **Follow-up response to Reviewer t5J4**
> >
> > Please find in this note our responses to your questions.
> >
> > 1. *MBR decoding uses BLEURT and is prohibitively expensive. Note, however, that more efficient alternatives exist. See implementation details discussed in [1], Section 4.3. Is there a reason to choose BLEURT? While efficiency at inference time is an advantage, MBR/QE finetuning is expensive at training time. It would be good to see some numbers to understand the training/inference time difference when compared to other existing approaches.*
> >
> > The two proposals in [1], Section 4.3 for increasing the efficiency of MBR decoding are to 1) use unique samples (deduping the candidate list) and 2) use non-neural utility functions (chrF++ and BLEU) or use COMET instead of BLEURT as the utility function. Both of these proposals increase efficiency at the cost of quality.
> >
> > Deduping (using unique samples) removes the implicit weighting of hypotheses by their model probability (counter to the intention of MBR) and hurts performance substantially based on internal results. Using chrF++, BLEU, and COMET as MBR utility functions all underperform using BLEURT, based on internal experiments which were performed to choose the best utility function. We hypothesize that COMET's separate encoding of the candidate and pseudo-reference translations, while more efficient, hurts quality. All of this said, comparing different utility functions is outside the scope of this paper, even though we agree that it is an interesting avenue of research.
> >
> > 2. *In the introduction you say that MBR decoding requires the generation of a large number of candidates. While this is true if you get unbiased samples from the model, it does not seem to be the case when you bias the distribution (e.g., using nucleus sampling). I suggest you see the discussions in [2] and [3] and comment. Also, see my comment above about [1].*
> >
> > It was shown in https://arxiv.org/pdf/2305.09860.pdf (Epsilon Sampling Rocks: Investigating Sampling Strategies for Minimum Bayes Risk Decoding for Machine Translation; Freitag et al., 2022) that epsilon sampling is the best sampling strategy for MBR decoding. In particular, it was shown that epsilon sampling outperforms nucleus sampling (according to human evaluation) when using a candidate size of 1024.
> >
> > Also see Table 3 and Table 4 in https://arxiv.org/pdf/2311.05263.pdf (Model-based Minimum Bayes Risk Decoding; Jinnai et al., 2023) for a comparison of MBR decoding using epsilon, top-k, nucleus, and ancestral sampling (with candidate size of 256), where epsilon sampling performs best for both English-German and Russian-English. (Also see Appendix C in the same paper, where it is shown that epsilon sampling outperforms all other sampling methods not only for machine translation, but also for text generation and image captioning tasks.)
> >
> > Moreover, note that in Figure 4 in the Appendix of our paper, we show that QE reranking performance monotonically improves with increasing candidate size (while, interestingly, this is not the case for QE finetuning). This is true even when using epsilon sampling which, like nucleus sampling, biases the distribution.
> >
> > 3. *Results use COMET-20 instead of more recent versions already available online such as COMET-22. Is there a reason for using the 2020 version?*
> >
> > We already report performance for both English-German and English-Japanese models on four diverse automatic metrics in Table 5 and Table 8, respectively. We chose COMET-20, rather than COMET-22, as our primary automatic evaluation metric since it is still the canonical metric used in the literature.
> >
> > As requested, we have also added COMET-22 scores to **Table 8** (for English-German) and **Table 11** (for English-Japanese). (We also added chrF as a representative lexical metric for completeness, as requested by another reviewer.)
> >
> > 4. *Human evaluation results are important and should not be in the appendix (Table 7). In fact, I think they should be more highlighted in the paper! It would also be interesting to see if they generalize for En-Ja. Can you explain the reasoning for using 9 professional translators for En-De instead of using fewer and evaluate En-Ja as well?*
> >
> > Human evaluation results have been moved to the main body of the paper. The human evaluation was only performed on English-German due to budget constraints and lack of availability of an English-Japanese rater pool within a reasonable timeframe.
> >
> > (To be continued in follow-up comment)

---

> > > ### Author Response · Authors · 2023-11-17
> > > **Follow-up response to Reviewer t5J4**
> > >
> > > 5. *What happens if you decode with reranking techniques (QE/MBR) using a model trained with your method, instead of using beam/greedy search? Even though this would make the method very inefficient at inference time, it would be interesting to see if it further boosts the performance.*
> > >
> > > As requested, we have added **Table 19** with MBR/QE decoding results for the MBR/QE-finetuned models, in comparison against the more efficient decoding methods of beam search, greedy decoding, and epsilon sampling. (In addition, we have also added the performance of not only QE-finetuned, but also MBR-finetuned, models to **Table 4**.)
> > >
> > > We see that after MBR finetuning, QE reranking works better than MBR decoding. Similarly, after QE finetuning, MBR decoding works better than QE reranking. This suggests that using the same decoding method both to generate the finetuning data and at inference time overfits to the metric used as the utility function. Also note that the gap between MBR/QE decoding and beam search/greedy decoding is small for the MBR/QE-finetuned models, and for Phases 2 and 3 of the QE-finetuned models, as well as Phase 3 of the MBR-finetuned model, beam search or greedy decoding is actually the top-performing decoding strategy.
> > >
> > > 6. *According to Table 3, both beam and greedy search work well. Does this mean that your method helps solve the beam search curse [4]? What happens when you increase the beam width?*
> > >
> > > Beam search with a small beam size (of 4, used here) should not be subject to the beam search curse. Moreover, the fact that greedy search performs on par with beam search certainly does not suggest that the beam search curse is solved with MBR/QE finetuning. If the beam search curse were solved, we should see that beam search with a beam size of 4 outperforms greedy, and this would motivate further experiments with a larger beam width. But that is not what we observe here.
> > > Since greedy decoding performs on par with beam search here, we are able to benefit from a very efficient decoding method for the MBR/QE-finetuned models at inference time.
> > >
> > > We hope the additions we have made to our paper allay your concerns and will be taken into account when assigning your final score. Please let us know if you have any further questions or if we can provide any additional clarifications to help finalize your assessment of our paper. Thank you!

---

> > > > ### Comment · Reviewer_t5J4 · 2023-11-21
> > > >
> > > > Thanks for your response and for updating the paper with some of my suggestions. My comment about the study being limited in terms of language coverage remains unchanged. In any case, I think you did a good job at addressing some of my concerns and updated my score accordingly.

---

> > > > > ### Author Response · Authors · 2023-11-22
> > > > > **Addition of experiments for German-English language pair**
> > > > >
> > > > > Thank you very much for taking the time to review our updates to the paper.
> > > > >
> > > > > To address your comment about the study being limited in terms of language coverage, we have completed Phase 1 experiments for an into-English language pair (German-English). We performed both MBR finetuning and QE finetuning using the (source side of the) WMT 2009-2019 test sets (30,695 examples total), and compared this against finetuning on the references and on beam translations.
> > > > >
> > > > > The trends align with those observed for the other language pairs in our study. Here, we observe that MBR finetuning and QE finetuning both improve performance relative to the base model, while beam finetuning does not. Moreover, QE finetuning outperforms finetuning on references (as also observed for English-Japanese). Please see the table of results below, sorted by Comet20 (and, incidentally, also by Comet22):
> > > > >
> > > > > | **de-en**       | BLEURT | BLEURT-X | ChrF  | Comet22 | Comet20 |
> > > > > | ------------------- | ------ | -------- | ----- | ------- | ------- |
> > > > > | Base model          | 72.96  | 78.03    | 76.55 | 84.17   | 51.77   |
> > > > > | Beam-finetuned      | 72.97  | 78.02    | 75.94 | 84.18   | 51.77   |
> > > > > | MBR-finetuned       | 73.62  | 78.45    | 76.74 | 84.38   | 53.31   |
> > > > > | Reference-finetuned | 73.64  | 78.76    | 76.00 | 84.67   | 54.15   |
> > > > > | QE-finetuned        | 73.86  | 78.99    | 76.54 | 84.81   | 55.10   |
> > > > >
> > > > > We hope the addition of the German-English experiments will help resolve your concerns about language coverage (and, in particular, about whether the techniques of MBR and QE finetuning generalize to into-English language pairs). We sincerely appreciate your careful and considerate review of our paper. Thank you again!

---

> > > > > > ### Comment · Reviewer_t5J4 · 2023-11-23
> > > > > >
> > > > > > Thank you for providing evidence that your method also works when translating into English. I think these results, when finished, should be added to the paper for completeness. I updated my initial review accordingly.

---

### Official Review · Reviewer_VTjn · 2023-11-03

**Soundness:** 3 good
**Presentation:** 3 good
**Contribution:** 3 good
**Rating:** 6
**Confidence:** 3

**Summary:**

This paper uses expensive decoding methods like MBR decoding and QE using neural metrics to generate data for finetuning. After fine-tuning, the model can be used with cheaper decoding strategies like beam search/greedy search and still retain the gains of the expensive decoding methods.
The paper explores different setups: finetuning on references, finetuning on MBR/QE decoded reference set, finetuning on MBR/QE decoded sampled monolingual data, finetuning on MBR/QE decoded samples from a very strong teacher (finetuned LLM) and show that finetuning on MBR/QE decoded monolingual data can get additional benefits over finetuning on references. They also show that finetuning using decoded data from a very strong teacher shows the best results.
The paper also does ablations around the effect of candidate size/source sentences selected/forward translation vs. backward translation while generating the finetuning data.
Finally the QE finetuned systems are evaluated against baselines in an MQM human evaluation to confirm the rankings produced by COMET-20

**Strengths:**

Strengths of the paper:

1. Sound experimental setup, clear description of the proposed method and showing its effectiveness
2. Important contribution for practitioners because currently MBR and QE decoding with neural metrics is computationally infeasible to be deployed in real time, the proposed approach can help bring some of those gains to production systems

**Weaknesses:**

Weaknesses of the paper:

1. The paper presents a section of the results from Palm2 finetuning and show large improvements. I think that is expected because it is a much stronger model. This set of experiments doesn't add too much value to the paper. The paper should also show the beam and MBR decoded results of the Palm2 model used for finetuning, so that the readers can understand show much of the teacher model performance can be transferred to the student via this finetuning.
2. After using MBR/QE finetuning, do we still see a difference between beam decoding and MBR/QE decoding? Table 3 summarizes the results using beam/greedy and sampling, but MBR/QE decoding results should also be included for completeness.
3. In Table 1, we see 2a perform better than 1c) and 2c) perform better than 1b). This is a bit suprising because I would have assumed that finetuning performance will be upper bounded by MBR/QE used during run-time. I couldn't find any comment on this comparison in the paper.
4. The paper does not have any numbers on lexical metrics like chrF. Would have been nice to include those numbers atleast in the Appendix, for interested readers

**Questions:**

Questions covered in the weakness section. No other specific questions.

---

> ### Author Response · Authors · 2023-11-17
> **Response to Reviewer VTjn**
>
> Dear Reviewer,
>
> Thank you for your thoughtful review of our paper. We appreciate that you recognize both the soundness and effectiveness of the proposed methods, as well as their importance for practitioners. We will seek to individually address each your outstanding concerns here.
>
> 1. *The paper presents a section of the results from Palm2 finetuning and show large improvements. I think that is expected because it is a much stronger model. This set of experiments doesn't add too much value to the paper. The paper should also show the beam and MBR decoded results of the Palm2 model used for finetuning, so that the readers can understand show much of the teacher model performance can be transferred to the student via this finetuning.*
>
> As requested, we have added the performance of the en-de and en-ja PaLM-2 QE-reranked, MBR-decoded, and greedy-decoded teacher models in **Table 10** and **Table 12** in the new revision, respectively. Note that for en-de, the student model is strong enough that MBR finetuning brings its performance almost on par with that of the PaLM-2 teacher model (student Comet20 = 63.62 vs teacher Comet20 = 63.76). Also note that across both language pairs, the MBR-decoded teacher is strongest. Moreover, even though the PaLM-2 QE-reranked teacher is weaker than the greedy-decoded teacher, PaLM-2-QE finetuning achieves much larger gains that PaLM-2-greedy finetuning (rows 3a versus 3c in Table 2). This suggests that QE finetuning is able to smooth out bad QE translations, e.g. due to utility function failure modes (while QE reranking at inference time is not robust to bad translations), allowing the student to exceed the performance of the teacher. In particular, the en-de PaLM-2-QE-finetuned student achieves a Comet20 score of 62.35, exceeding its teacher's score of 60.86.
>
> 2. *After using MBR/QE finetuning, do we still see a difference between beam decoding and MBR/QE decoding?*
>
> As requested, we have added **Table 19** with MBR/QE decoding results for the MBR/QE-finetuned models, in comparison against the more efficient decoding methods of beam search, greedy decoding, and epsilon sampling. (In addition, we have also added the performance of not only QE-finetuned, but also MBR-finetuned, models to **Table 4**.)
>
> We see that after MBR finetuning, QE reranking works better than MBR decoding. Similarly, after QE finetuning, MBR decoding works better than QE reranking. This suggests that using the same decoding method both to generate the finetuning data and at inference time overfits to the metric used as the utility function. Also note that the gap between MBR/QE decoding and beam search/greedy decoding is small for the MBR/QE-finetuned models, and for Phases 2 and 3 of the QE-finetuned models, as well as Phase 3 of the MBR-finetuned model, beam search or greedy decoding is actually the top-performing decoding strategy.
>
> 3. *In Table 1, we see 2a perform better than 1c) and 2c) perform better than 1b). This is a bit suprising because I would have assumed that finetuning performance will be upper bounded by MBR/QE used during run-time. I couldn't find any comment on this comparison in the paper.*
>
> MBR decoding and QE reranking can produce translations with high utility score but low quality, due to honeypots/failure modes of their utility functions. While this directly degrades performance of the teacher models at inference time, the MBR and QE-finetuned students are able to smooth out these bad translations. This explains why student performance is not upper-bounded by performance of the teacher and can, in fact, exceed it.
>
> This hypothesis is supported by work from Amrhein and Sennrich, 2022 (https://arxiv.org/pdf/2202.05148.pdf) which shows that, despite the strong performance of MBR decoding and QE reranking on average, these decoding methods suffer from failure modes which can be attributed to underlying problems with the utility function. They performed the study using Comet20 as the utility function, and showed that this metric is not sensitive enough to discrepancies in numbers and named entities. MBR and QE finetuning, on the other hand, would not necessarily be as susceptible to these same failure modes since, even though the model would be exposed to these extreme examples, its representation would be smoothed out over the course of training.
>
> 4. *The paper does not have any numbers on lexical metrics like chrF. Would have been nice to include those numbers atleast in the Appendix, for interested readers.*
>
> As requested, we have added performance on chrF to **Table 8** (for English-German) and **Table 11** (for English-Japanese) in the Appendix.
>
> We hope the additions we have made to our paper allay your concerns and will be taken into account when assigning your final score. Please let us know if you have any further questions or if we can provide any additional clarifications to help finalize your assessment of our paper.

---

### Official Review · Reviewer_2DMZ · 2023-11-06

**Soundness:** 3 good
**Presentation:** 3 good
**Contribution:** 2 fair
**Rating:** 6
**Confidence:** 4

**Summary:**

The paper proposes two new training methods called MBR finetuning and QE finetuning to distill the benefits of high-quality decoding methods like Minimum Bayes Risk (MBR) decoding and Quality Estimation (QE) reranking into the model weights, while avoiding expensive inference costs. The results demonstrates their effectiveness for machine translation including exceeding human references

**Strengths:**

1. MBR and QE produce superior quality but are too expensive for inference. The finetuning approach provides a way to achieve most of their benefits without sacrificing efficiency. This could enable deploying higher performance models in real applications.
2. The proposed methods are clearly explained and technically sound. The experiments are comprehensive and rigorous, spanning various metrics, domains, language pairs, and resource settings. The results consistently validate the effectiveness of MBR and QE finetuning over strong baselines.

**Weaknesses:**

1.	The novelty of the method is unclear. Using high-quality pseudo data generated by better decoding methods is a common technique in neural machine translation. This paper does not seem to provide new insights beyond what is already known.
2.	While this paper reports significant gains with MBR and QE finetuning, the reasons for the improvements are not well analyzed. In some cases, QE finetuning performs better, while in others, MBR finetuning is superior. More details and an in-depth analysis of the advantages and limitations of each finetuning method would be expected.

**Questions:**

1.	Please report the performance of the PaLM-2 Bison model.
2.	For English-German, MBR finetuning seems to outperform QE finetuning. Why was only QE finetuning evaluated for English-Japanese instead of also evaluating MBR?
3.	Can you report the training time and compute requirements for the QE and MBR finetuning experiments? Providing the concrete training costs for comparison would be helpful.

---

> ### Author Response · Authors · 2023-11-17
> **Response to Reviewer 2DMZ**
>
> Dear Reviewer,
>
> Thank you for your constructive and thoughtful review of our paper. We appreciate your recognition of the soundness and practical importance of the proposed methods. We also appreciate your taking the time to carefully review our experiments, and your acknowledgement that they are comprehensive and rigorous. To allay your outstanding concerns, we will respond individually here to each of the weaknesses and questions that you raised.
>
> 1. *The novelty of the method is unclear. Using high-quality pseudo data generated by better decoding methods is a common technique in neural machine translation. This paper does not seem to provide new insights beyond what is already known.*
>
> The review states that "using high-quality pseudo data generated by better decoding methods is a common technique in neural machine translation", but does not cite any papers to support this claim. We are not aware of any such work, so if calling into question the novelty of MBR and QE finetuning, please kindly provide evidence that "this paper does not seem to provide new insights beyond what is already known". Note: We show that self-finetuning using beam search output does **not** improve quality, so the decoding strategy used to generate the finetuning data is crucial to the success of this method.
>
> 2. *While this paper reports significant gains with MBR and QE finetuning, the reasons for the improvements are not well analyzed. In some cases, QE finetuning performs better, while in others, MBR finetuning is superior. More details and an in-depth analysis of the advantages and limitations of each finetuning method would be expected.*
>
> This paper does provide an explanation for the improvements observed with MBR and QE finetuning in paragraph 2 of the Discussion section (Section 7), and also provides a comparison of MBR versus QE finetuning in paragraph 3. Both paragraphs are pasted below for reference:
>
> 1. **Explanation for improvements (Section 7, paragraph 2)**: "Given the effectiveness of an external teacher model (which differs in architecture, training data, etc. from the student), the success of these finetuning methods can likely not be primarily explained by mitigation of the label exposure bias problem (Schmidt, 2019), since the data generated from an external teacher is less similar to the data the model has already seen (during previous training stages) than self-generated MBR or QE data. Instead, the effectiveness of MBR and QE finetuning seemingly can be attributed primarily to the high quality of the translations used as finetuning data."
> * Also note that the experiments were designed so that each of the 3 phases answered a different question regarding the effectiveness of MBR and QE finetuning (see **Section 5.4**). For example, by comparing Phase 1 and Phase 2, we show that using a larger monolingual corpus to generate the MBR and QE finetuning data improves performance, and by comparing Phase 2 and Phase 3, we show that using a stronger teacher model also improves performance.
> * Also see **Appendix D** for extensive ablation studies which isolate the effect of various source-side and target-side variables on the performance of MBR and QE-finetuned models. For instance, we show that QE finetuning on forward-translated data is more effective than QE finetuning on backtranslated data (where the QE translations are on the source, rather than target, side). All of these ablations serve to understand the reasons for the observed improvements.
> 2. **Comparison of MBR vs QE finetuning (Section 7, paragraph 3)**: "When comparing MBR against QE finetuning, we see that QE tends to perform at least as well as MBR finetuning using a self-teacher, while MBR outperforms QE finetuning using the LLM teacher. We hypothesize that the sampling translations from the self-teacher do not closely approximate the true reference distribution, so using them as pseudo-references does not help (or even hurts), relative to using a reference-free (QE) metric. For higher-quality models (e.g. PaLM-2), the candidate translations are good enough approximations to references that they provide a useful signal."
> * We also mention in the introduction that generating data via QE reranking is much more efficient than generating data via MBR decoding: "Despite its quality advantages, the slow inference speed of MBR decoding remains a limitation even when generating distillation data". In terms of computational cost, this is the key limitation of MBR finetuning. So the limitations of MBR relative to QE finetuning both in terms of quality (MBR finetuning requires a stronger teacher) and computational cost are included in the paper. Also see **Table 7** (added to the paper based on your comment #3).
>
> If 1. and 2. above do not satisfy the requirement for 1) an analysis of the reasons for the improvements due to MBR and QE finetuning and 2) a comparison of the advantages and limitations of MBR vs QE finetuning, please do clarify what is missing.

---

> > ### Author Response · Authors · 2023-11-17
> > **Follow-up response to Reviewer 2DMZ**
> >
> > Please find in this note our responses to your questions:
> >
> > 1. *Please report the performance of the PaLM-2 Bison model.*
> >
> > As requested, we have added the performance of the English-German and English-Japanese PaLM-2 greedy-decoded, QE-reranked, and MBR-decoded teacher models in **Table 10** and **Table 12** in the new revision, respectively. Note that for English-German, the student model is strong enough that MBR finetuning brings its performance almost on par with that of the teacher model (student Comet20 = 63.62 vs teacher Comet20 = 63.76). Also note that across both language pairs, the MBR-decoded teacher is strongest. Moreover, even though the PaLM-2 QE-reranked teacher is weaker than the greedy-decoded teacher, PaLM-2-QE finetuning achieves much larger gains that PaLM-2-greedy finetuning (rows 3a versus 3b in Table 2). This suggests that QE finetuning is able to smooth out bad QE translations, e.g. due to utility function honeypots/failure modes (while QE reranking at inference time is not robust to bad translations), allowing the student to exceed the performance of the teacher. In particular, the English-German PaLM-2-QE-finetuned student achieves a Comet20 score of 62.35, exceeding its teacher's score of 60.86.
> >
> > 2. *For English-German, MBR finetuning seems to outperform QE finetuning. Why was only QE finetuning evaluated for English-Japanese instead of also evaluating MBR?*
> >
> > First, to address the claim that "MBR finetuning seems to outperform QE finetuning": For English-German, MBR finetuning only outperforms QE finetuning for the stronger teacher models (self-teacher finetuned on references and PaLM-2 teacher), but QE finetuning outperforms MBR finetuning for the weaker teacher (self-teacher base model). This is brought up in the Discussion section (Section 7, paragraph 3). Also note that MBR and QE finetuning are not directly comparable since they use different utility functions. So we decided to only perform QE finetuning for English-Japanese, since we already ran experiments for both MBR and QE finetuning for English-German, and since QE data generation is much more efficient than MBR data generation. We were already able to achieve large gains with QE finetuning for English-Japanese, without having to resort to more expensive data generation, which is a win both in terms of quality and computational complexity.
> >
> > 3. *Can you report the training time and compute requirements for the QE and MBR finetuning experiments? Providing the concrete training costs for comparison would be helpful.*
> >
> > As requested, we have added **Table 7** (**Appendix B**) in our paper revision, which shows the time and compute requirements for performing MBR-BLEURT and QE-MetricX-XXL scoring, across all three experiment phases and across both language pairs. We use TPUs for BLEURT and MetricX-XXL scoring, as described in https://dl.acm.org/doi/fullHtml/10.1145/3360307. (The TPUv4 hardware is a faster and newer generation than TPUv3.) MBR scoring takes about 7.65 TPUv4 minutes/source sentence, while QE scoring takes about 4.2 TPUv3 seconds/source sentence. So even when using faster hardware for MBR, QE scoring is 109x faster.
> >
> > As for training time and costs, finetuning was performed using 64 TPUv3s, with a rate of ~0.2 steps per second using a batch size of 32, so training for one epoch on the Common Crawl dataset (from Phases 2 and 3) took ~8.7 hours. The model was also evaluated on the validation set at every step, since the performance of MBR and QE finetuned models often peaked after only a few steps.
> >
> > We hope the additions we have made to our paper allay your concerns. Please let us know if you have any further questions or if we can provide any additional clarifications to help finalize your assessment and rating of our paper.

---

> > > ### Comment · Reviewer_2DMZ · 2023-11-23
> > >
> > > Thank you for your thoughtful response addressing my concerns and questions. While I still think that using complex decoding methods to generate higher quality pseudo data is unsurprising, I appreciate the revisions made to complete this paper with more extensive results presented. Considering these additional experiments and analysis, I update my score accordingly.

---

### Official Review · Reviewer_5sbu · 2023-11-07

**Soundness:** 3 good
**Presentation:** 3 good
**Contribution:** 3 good
**Rating:** 6
**Confidence:** 4

**Summary:**

The paper describes approaches to training NMT models that use different decoding methods than beam search, namely, QE reranking and MBR decoding. Using such decoding methods in inference can be quite expensive computationally but using them in training time allows for maintaining inference time efficiency while leveraging their mitigation of model-perplexity-vs-quality. This approach can be seen as an alternative to aligning MT models with human translation preferences, given by the QE or utility function used in MBR. Furthermore, using either QE reranking or MBR decoding allows for exploring monolingual data augmentation for training as neither require references for scoring translation hypotheses. Automatic and manual evaluation results indicate that QE-finetuned systems outperform the reference-finetuned baseline, and QE finetuning using a PaLM-2 teacher outperforms using the self-teacher model.

**Strengths:**

The paper is a good contribution towards aligning MT models with human preferences as given by reference-less or reference-based metrics (QE reranking and the utility function used for MBR, respectively). It allows leveraging monolingual data relevant to the MT model application setting for improving its translation quality. Automatic evaluation is consistent with manual evaluation and the experimental setting is sound. This is an interesting alternative to reinforcement learning models that have shown to be quite unstable in MT settings. The approaches shown here are also generalizable to other natural generation tasks though no experiments have been devised to show that.

**Weaknesses:**

The paper is quite dense at some parts, mainly at the experimental setting. It takes a few passes on those sections to completely understand the results and the settings but it is clear. The QE approach is not really reproducible as the QE model is not publicly available. Furthermore, the best teacher model seems to be a PaLM2-Bison that is also not publicly available.

**Questions:**

* Is there any description of MetricX-XXL-QE in another paper? I could not find a paper describing except a paragraph in the Findings paper. How many languages does it support? This seems to be a very big QE model, 30B parameters.

* How much time is added to the training process when using the MBR and QE reranking? Maybe a table illustrating the added wallclock time would be interesting to contrast with the gains in performance.

* How much does the data used to train the QE model and the utility function used in MBR affect the final results for specific domains? Do yo have any insights on this? For example, for English-German, it seems most of the MT training data is similar to the data used to train the QE and BLEURT models. What if this is not the case? Do you observe the same improvements or does something change in the translation quality of the best models?

---

> ### Author Response · Authors · 2023-11-17
> **Response to Reviewer 5sbu**
>
> Dear Reviewer,
>
> Thank you for your careful and thoughtful review of our paper. We observed that you lowered our score from 8 to 6 on November 14, and our hope is that we can address all of your outstanding concerns here.
>
> First, to address your concerns brought up in the initial revision of the review:
>
> Acknowledged that the experimental settings section of the paper is "dense". If there is anything unclear about the setup, please do specify and we will be happy to make changes to clarify. We have added **Table 6** in the Appendix (with a reference to this table in Section 5.4) with a summary of the experimental settings (in particular, the datasets used for each experimental phase). We hope this will make the experimental settings section more easily digestible with a single pass.
>
> When the score was revised down from 8 to 6, we observed that the diff of the revised review with respect to the original review was the following sentence: *"Some missing points of comparison are approaches that leverage feedback for MT."*
>
> Could you kindly clarify which approaches that leverage feedback for MT are missing? Since no citations were provided, we are unable to comment on this further without additional information. Also, please note that leveraging feedback for MT is not the topic of this paper, so such a comparison may be out of scope.
>
> We will now respond individually to each of your questions:
> 1. *Is there any description of MetricX-XXL-QE in another paper? I could not find a paper describing except a paragraph in the Findings paper. How many languages does it support? This seems to be a very big QE model, 30B parameters.*
>
> The description of the reference-based counterpart to MetricX-XXL-QE (trained on the same human judgements data and with the same model architecture) can be found in https://www2.statmt.org/wmt23/pdf/2023.wmt-1.63.pdf (MetricX-23: The Google Submission to the WMT 2023 Metrics Shared Task; Juraska et al., 2023). The only difference between the original MetricX model and the QE version we use is that the QE model input is constructed as the concatenation of the source segment, rather than the reference, with the candidate translation (as described in paragraph 3 of Section 2.3 in our paper). This model has a mT5-XXL backbone with 13B parameters.
>
> As shown in the WMT'23 Metrics Shared Task Report (https://www2.statmt.org/wmt23/pdf/2023.wmt-1.51.pdf), MetricX generalizes well to language pairs that it didn't see during training, including low-resource language pairs. See, for example, Hebrew-English performance in Table 8.
>
> 2. *How much time is added to the training process when using the MBR and QE reranking? Maybe a table illustrating the added wallclock time would be interesting to contrast with the gains in performance.*
>
> To address your question, we have added **Table 7** (**Appendix B**) in our paper revision, which shows the time and compute requirements for performing MBR-BLEURT and QE-MetricX-XXL scoring, across all three experiment phases and across both language pairs. We use TPUs for BLEURT and MetricX-XXL scoring, as described in https://dl.acm.org/doi/fullHtml/10.1145/3360307. (The TPUv4 hardware is a faster and newer generation than TPUv3.) MBR scoring takes about 7.65 TPUv4 minutes/source sentence, while QE scoring takes about 4.2 TPUv3 seconds/source sentence. So even when using faster hardware for MBR, QE scoring is 109x faster. Please do let us know if you have additional questions about this new table.
>
> 3. *How much does the data used to train the QE model and the utility function used in MBR affect the final results for specific domains? Do yo have any insights on this? For example, for English-German, it seems most of the MT training data is similar to the data used to train the QE and BLEURT models. What if this is not the case? Do you observe the same improvements or does something change in the translation quality of the best models?*
>
> In the WMT'22 Metrics Shared Task (https://aclanthology.org/2022.wmt-1.2.pdf), the metrics (including BLEURT and MetricX-XXL) were evaluated on various domains, and they performed well even on the domains that they were not finetuned on (see **Figure 2**).
>
> Moreover, just as the utility functions generalize across domains, the same is true for the MBR-finetuned and QE-finetuned models which use these utility functions, as shown in **Table 9** in our paper. Also note that while the Phase 1 models were finetuned on past WMT test sets (with the source side partially overlapping the data used to train the utility functions), the Phase 2 and Phase 3 models were finetuned on Common Crawl data, and we do see that finetuning on Common Crawl improves generalization across domains.
>
> We hope the additions we have made to our paper allay your concerns. Please let us know if you have any further questions or if we can provide any additional clarifications to help finalize your assessment and rating of our paper.

---

> > ### Comment · Reviewer_5sbu · 2023-11-23
> >
> > Thank you for the answers. I've seen the discussion about comparison with other methods that leverage feedback and agree that this is somewhat tangential. Regarding the other questions, thank you for the the work, I think the paper improved after this discussion period. I updated the scores accordingly.

---

### Author Response · Authors · 2023-11-17
**General response**

We sincerely thank all the reviewers for their feedback and constructive comments. We are pleased that the reviewers unanimously appreciate the effectiveness of the proposed methods of MBR and QE finetuning, the sound setup and rigorous experiments, the clear presentation, and the important practical applications of these methods to reduce inference-time costs while maintaining quality gains.
We have updated our manuscript to reflect reviewers' comments (all revisions are highlighted in red in the new PDF). The major updates are summarized as follows:
1. **Table 6**: Summary of experimental setup in table format (R#1)
2. **Table 7**: Time and compute costs of generating the MBR and QE data (R#1, R#2)
3. **Table 10** and **Table 12**: Performance of PaLM-2 MBR-decoded and QE-reranked teacher models (R#2, R#3)
4. **Table 8** and **Table 11**: Add chrF (R#3) and Comet22 (R#4) as additional evaluation metrics.
5. **Table 19**: Performance of MBR-finetuned and QE-finetuned models using MBR/QE decoding, as compared against more efficient decoding methods (R#3, R#4)

Below, we address all weaknesses and questions raised by each reviewer individually. We again thank each reviewer sincerely for the time and thoughtful consideration put into our reviews. Should there be a need for further clarification to assist in advancing our score, please do not hesitate to let us know.

---

### Meta-Review · Area_Chair_Yd7z · 2023-12-14

**Metareview:**

This paper proposes a training-time distillation approach for MBR decoding, avoiding the expensive runtime of MBR at decoding time. There were some concerns from the reviewer about this paper which have been mostly resolved in the rebuttal phase -- all reviewers feel the paper is marginally above the acceptance threshold. The new version of the model expands the language coverage. There were also some concerns about reproducibility, since some of the models used in this paper are not publicly available (the QE model and the best teacher model). Overall, while the proposed approach is relatively incremental, I I find this a solid paper and the empirical work is rigorous and comprehensive, therefore I recommend acceptance.

**Justification For Why Not Higher Score:**

Somewhat incremental and not reproducible.

**Justification For Why Not Lower Score:**

The proposed approach seems novel and the empirical part is comprehensive.

---

### Decision · Program_Chairs · 2024-01-16

Accept (poster)